# Analysis of Whole Genome Resequencing Datasets from a Worldwide Sample of Sheep Breeds to Identify Potential Causal Mutations Influencing Milk Composition Traits

**DOI:** 10.3390/ani10091542

**Published:** 2020-09-01

**Authors:** Héctor Marina, Beatriz Gutiérrez-Gil, Cristina Esteban-Blanco, Aroa Suárez-Vega, Rocío Pelayo, Juan José Arranz

**Affiliations:** Departamento de Producción Animal, Facultad de Veterinaria, Universidad de León, Campus de Vegazana s/n, 24071 León, Spain; hmarg@unileon.es (H.M.); bgutg@unileon.es (B.G.-G.); cestb@unileon.es (C.E.-B.); asuav@unileon.es (A.S.-V.); rpelg@unileon.es (R.P.)

**Keywords:** dairy sheep, milk production, genetic polymorphism, ruminants, whole genome sequence

## Abstract

**Simple Summary:**

Most of the production of sheep’s milk is used in the manufacture of mature cheeses. The milk composition has a strong influence on the technological and organoleptic properties of dairy products. Several genetic polymorphisms have been related to variations of milk protein and milk fatty acid content. The use of whole genome resequencing (WGR) has encouraged the discovery of polymorphisms in the sheep genome. Exploiting information derived from a large number of sheep WGR datasets, this study aimed to evaluate the variability of 24 candidate genes involved in physiological pathways related to milk production. The genetic variants highlighted by this work have a potential influence on the function of the protein encoded by the candidate genes. The relevance of sheep milk composition on the cheese-making industry enhances the potential interest of the present study as the variants highlighted herein could be considered to increase the efficiency of breeding programs currently applied to dairy sheep populations. Further studies would be needed to understand the role of these genetic variants on milk production traits.

**Abstract:**

Different studies have shown that polymorphisms in the sequence of genes coding for the milk proteins and milk fatty acids are associated with milk composition traits as well as with cheese-making traits. However, the lack of coincident results across sheep populations has prevented the use of this information in sheep breeding programs. The main objective of this study was to exploit the information derived from a total of 175 whole genome resequencing (WGR) datasets from 43 domestic sheep breeds and three wild sheep to evaluate the genetic diversity of 24 candidate genes for milk composition and identify genetic variants with a potential phenotypic effect. The functional annotation of the identified variants highlighted five single nucleotide polymorphisms (SNPs) predicted to have a high impact on the protein function and 42 missense SNPs with a putative deleterious effect. When comparing the allelic frequencies at these 47 polymorphisms with relevant functional effects between the genomes of Assaf and Churra sheep breeds, two missense deleterious variants were identified as potential markers associated to the milk composition differences found between the Churra and Assaf: *XDH:92215727C>T* and *LALBA:137390760T>C*. Future research is required to confirm the effect of the potential functionally relevant variants identified in the present study on milk composition and cheese-making traits.

## 1. Introduction

Selection for specialized production traits and adaptation to a wide range of environments has involved changes within the genome of modern sheep breeds. Sheep specialization for milk production began 4000–5000 years ago [1]. In the last few hundred years, the development of the different sheep breeds and the use of quantitative genetics methodology have resulted in the establishment of dairy sheep breeds, some of which show a high level of specialization for milk production [2]. Due to the higher total solid content of sheep milk compared to that of other dairy species, the main purpose of dairy sheep production is the manufacturing of high-quality cheese. Sheep milk is an excellent supplier of proteins, energy, fat, minerals, and vitamins [3,4]. Milk protein and fat content as well as the total solids content are directly correlated with the cheese yield trait [5,6]. Therefore, genetic selection programs in dairy sheep take into account not only milk yield, but also milk composition traits.

Since about the 1980s, several studies have tried to identify genetic markers that could be used to improve the efficiency of selection for these traits through the candidate gene approach, focusing on the study of milk protein genes and genes related to milk fat content. The initial studies on this field were focused on the study of polymorphisms in the genes coding for the caseins (Cn), which represent more than 95% of the proteins contained in sheep milk [7]. Sheep milk has four types of caseins, αs1-Cn, αs2-Cn, β-Cn, and κ-Cn, and their coding genes (*CSN1S1*, *CSN1S*, *CSN2*, and *CSN3*, respectively) are clustered on a region of ovine chromosome 6 (OAR6) spanning 250 kb [8,9]. Polymorphisms in the coding regions of the casein genes have been found to be associated with effects on milk yield, milk protein, and fat percentages [10,11]. Genetic variants in these genes have also been related to milk coagulation properties such as rennet coagulation time and curd firmness [8]. For many polymorphisms in the genes coding for milk sheep caseins, significant associations have been reported for quantitative and qualitative milk parameters [12]. Other studies have also focused on the genetic variants of the genes coding for the sheep milk whey proteins, α-Lactalbumin and β-Lactoglobulin, which account for 17–22% of the total milk proteins in sheep milk [13]. These whey proteins are encoded by two genes located on OAR3, the *LALBA* gene and the *LGB* gene (annotated as *PAEP* in the sheep reference genome Oar_v3.1), both located on regions that have been associated with milk performance traits in dairy sheep [14,15]. In this sense, it is worth mentioning that a missense variant in the coding region of the *LALBA* gene (p.Val27Ala) was identified through a GWAS study as the causal mutation of a previously identified QTL effect influencing milk protein and milk fat content in Spanish Churra sheep [16]. Another study carried out by Corral et al. [10] related a missense variant within the *PAEP* gene (p.Tyr38His) to higher protein and fat percentages in Merino ewes. On the other hand, further important factors that influence the manufacturing properties and organoleptic quality of dairy products are the milk fat content and the milk fatty-acid composition pattern. Fat content is a highly variable component of milk, dependent on breed, genotype, and diet [17]. This can be explained because lipid metabolism in the mammary gland is a complex process that involves a large number of genes and their interactions [18]. Therefore, researchers have also applied the candidate gene approach to identify polymorphisms in genes that could influence the fat composition of sheep milk such as *ACACA*, *CSN1S1*, *CSN2*, *FASN*, and *LALBA* [11,15,19,20]. 

Overall, these studies suggest that polymorphisms in the milk protein genes and the genes coding for fatty-acid syntheses in the mammary gland appear to have a direct effect on milk composition traits, and indirectly on cheese-making properties and cheese yield-related traits. However, up to date, the controversial results obtained in the different sheep breeds have prevented the use of this molecular information in the practice of milk selection programs [3]. Hence, information on the genetic diversity of these genes across different sheep populations could be of interest. For instance, evidence of the presence of selection signatures around these genes could suggest a potential direct effect of the corresponding gene on phenotypes of interest for the dairy sheep industry. 

The Sheep HapMap project was a first attempt to decipher the genetic variability of the sheep genome across a large sample of worldwide ovine breeds through the genotyping of a medium-density SNP-chip (50K-chip) [2]. This study did not reveal any significant selection signal in the genomic regions harboring the candidate genes mentioned. As an extension of the Sheep HapMap project, the International Sheep Genomics Consortium (ISGC) initiated the “Ovis aries diversity study” in 2012, which included the whole genome sequences of a subset of samples from the Sheep HapMap project, a total of 75 sheep from 43 breed groups, and two wild species from around the world. The whole genome resequencing (WGR) datasets generated in this project are publicly accessible in the Sequence Read Archive (SRA) [PRJNA160933]. 

Considering the availability of these WGR datasets, the aim of this study was focused on assessing the genetic diversity of candidate genes for milk production/composition traits across a large number of breeds. Considering the limited number of available WGR datasets corresponding to specialized dairy breeds in the repositories, we analyzed here an additional dataset of 71 individual genomes including samples from two dairy breeds with different specialization level: Spanish Churra (*n* = 46) and Spanish Assaf (*n* = 19). These two breeds significantly differ in their dairy specialization level as the Assaf milk yield (400 kg; lactation normalized to 150 days) is more than double the milk yield in Churra (117 kg; lactation normalized to 120 days), whereas Churra milk has higher fat and protein contents and also increased cheese yields than the Assaf milk [21].

The analysis presented here for a total of 175 WGR datasets from a worldwide sample of sheep breeds has allowed us to evaluate the genetic diversity for a list of 24 milk composition candidate genes and estimate the relationship between domestic sheep breeds and their Asian mouflon ancestor (*Ovis orientalis*) based on the genome and candidate regions’ genetic variability. In addition, this work exploited the value of WGR datasets to identify potential functional variants that could underlie phenotypic variation for traits of economic interest in dairy sheep.

## 2. Materials and Methods

### 2.1. Whole Genome Resequencing (WGR) Datasets

This study involved the analysis of 151 WGR datasets from 43 domestic sheep breeds and 24 WGR datasets of three wild sheep (*O. canadiensis*, *O. dalli*, and *O. orientalis*). The detailed description and origin of the total 175 analyzed datasets are provided inAppendix A. 

Briefly, most of the domestic sheep analyzed WGR datasets were publicly available from two different projects at the Sequence Read Archive (SRA) repository. A total of 70 domestic sheep datasets were obtained from the project “Ovis aries diversity study” (PRJNA160933) developed by the International Sheep Genomics Consortium, which includes a representative group of samples previously analyzed within the HapMap project [2]. Another 10 WGR datasets of Australian Merino were obtained from the project “SheepCRC whole genome sequencing project” (PRJNA325682) developed by the Sheep Commonwealth Government’s Cooperative Research Center. This project has recently been extended with additional samples, but those analyzed here are the 10 initially available. In addition, we included in our study 71 WGR datasets generated by our research group for four Spanish domestic breeds [Spanish Assaf (*n* = 19), Churra (*n* = 46), Segureña (*n* = 2), and Spanish Merino (*n* = 4) (WGR datasets available under request through the corresponding agreement framework between authors and the particular sheep breeders’ associations). All WGR datasets generated by our group were produced using the paired-end Illumina technology (Illumina HiSeq 2000 and Hiseq 2500 sequencers). Among this group of Spanish breeds, Assaf sheep shows the highest specialization for milk production [22], whereas Churra is an autochthonous double aptitude breed classically exploited for milk production and suckling lamb meat [23]. Although the Assaf breed has a higher milk yield, Churra sheep have higher protein (5.98%) and fat (7.12%) contents in milk than Assaf sheep (4.03 and 5.39%, respectively) [6,24]. This explains that Churra sheep milk shows a higher cheese yield and a better aptitude for the production of mature cheeses than milk from Assaf sheep [25,26]. A recent study of our research group reported that Assaf milk shows a slow-coagulation during the cheese-making process [5], whereas Churra milk appears to have better coagulation properties and a higher cheese yield [6].

In addition, this study included additional WGR datasets available at the SRA repository for a total of 19 Asian mouflon (*Ovis orientalis*) generated by the NEXTGEN project (PRJEB3139) and two wild sheep breeds (*Ovis canadiensis*, *n* = 2 and *Ovis dalli*, *n* = 3) from the “Ovis aries diversity study” (PRJNA160933). The geographic origin of breeds included in this study is represented in Figure 1. The breeds of the domestic samples were drawn from Asia (7), Africa (4), America (4), Australia (2), British (5), Central Europe (3), Northern Europe (3), Southwest Europe (7), and Southwest Asia (9). On the other hand, the wild sheep species here considered belonged to America (2) and Southwest Asia (1). The origin of the ancestor mouflon samples, all from the Iran region, is also indicated in Figure 1.

### 2.2. Candidate Genes Considered

Genes considered in the analysis comprised of (i) the milk protein genes including the casein genes (*CSN1S1*, *CSN1S*, *CSN2*, and *CSN3*, respectively), and the milk whey proteins (*LALBA* and *PAEP* genes). A group of (ii) 17 other candidate genes related to milk fatty acid metabolism in the mammary gland [18] in concordance with a previous study performed by our research group assessed the expression of candidate milk genes in the milk sheep transcriptome [21]. The selected milk-fat composition candidates can be grouped following lipid metabolism processes: fatty acid synthesis and desaturation (*ACACA*, *FASN* and *SCD*), lipid droplet formation (*BTN1A1*, *XDH*), fatty acid activation and intracellular transport (*ACSL1*, *ACSS2*, *DBI* and *FABP3*), acetate triacylglycerol synthesis (*GPAM*, *DGAT1* and *LPIN1*), fatty acid import into cells (*CEL*, *LPL* and *VLDLR*), and other genes related with lipids metabolism such as *FABP4, PLIN2*, and *SLC27A6* [18,21]. The genome coordinates corresponding to the selected candidate genes according to the sheep reference genome Oar_v.3.1 were extracted with the BioMart tool of Ensembl [27]. The complete list of studied candidate genes and their genomic coordinates are detailed in Appendix A.

### 2.3. WGR Bioinformatics Analysis

Samples obtained from the SRA public repository were converted from SRA format to FASTQ format with the SRA-Toolkit software (available from http://www.ncbi.nlm.nih.gov/Traces/sra). After this transformation, all WGR datasets were analyzed through the same pipeline, which involved the following steps: (1) quality of the raw paired-end reads was assessed with the FastQC program [28]; (2) good quality reads were filtered with Trimmomatic [29] using specific filter parameters for paired-end samples (-phred33, LEADING:5, TRAILING:5 SLIDINGWINDOW:4:20, MINLEN:36 ILLUMINACLIP: Trimmomatic-0.33/adapters/TruSeq 3-PE.fa:2:30:10); (3) samples were aligned against the ovine reference genome assembly v.3.1 (Oar_v3.1 [27]) with the program Burrows-Wheeler Aligner (BWA) [30] using the algorithm of maximal exact matches (mem); (4) data manipulation and statistical analyses were performed with three programs: SAMtools [31] was used to convert SAM files into BAM files and to remove non-mapped and improperly-pair reads and obtain alignment statistics, Picard [32] was used to sort reads, mark duplicate reads and index building, and Genome Analysis Toolkit v4.0 (GATK) [33] was used to perform a base quality score re-calibration; (5) identification of genetic variants was carried out considering all 175 WGR datasets together with GATK v4.0 (University of Texas at Austin, Austin, TX, USA) [33] using the haplotypecaller tool and default parameters following GATK Best Practices recommendations; (6) filtering of the variants identified was performed with the program snpSIFT [34] in order to remove low quality variants (DP > 10 & QUAL > 30 & MQ > 30 & QD > 5 & FS < 60); (7) the BCFtools utilities (Genome Research Ltd., Cambridgeshire, UK) [31] were used to add the identifier code for each known variant, using a VCF file (release91) downloaded from Ensembl database as a reference [27]; and (8) variants located on candidate genes were extracted from all datasets with the SnpEff software (Pablo Cingolani, Boston, MA, USA) [34], using the –fi option along with a bed file providing information about the coordinates of each gene according to the sheep reference genome (Appendix A).

### 2.4. Genetic Diversity and Phylogenetic Analyses

Nucleotide diversity (π) was estimated in 20 kb genomic bins with a 10 kb step in both wild and domestic sheep genome collections, following Naval-Sanchez et al. [35], through the VCFtools software [36] and considering single nucleotide polymorphism (SNP) variants identified across the whole genome. Genomic bins with fewer than 20 SNP were excluded. The same approach was used to estimate π for the genomic regions harboring the candidate genes under study.

The estimation of evolutionary divergence among the samples studied was measured using the maximum composite likelihood method [37] through the 1% of the total variants detected across the genome among wild and domestic sheep, in order to construct the evolutionary divergence distance matrix among the samples. Results were drawn using the Neighbor-joining method with the MEGA X [38] and ggtree [39] software. Furthermore, evolutionary divergence distances, considering only the variants found in the candidate regions were estimated using the same pipeline. The reliability of phylogenetic tests was estimated through a bootstrapping method [40] using 500 bootstrap replicates [40] with MEGA X [38].

### 2.5. Functional Annotation and Site Frequency Study for Candidate Gene Variants

For the variants identified at the milk composition candidate genes, annotation and prediction of its effect were carried out by two software packages: (i) SnpEff [34] to predict the impact of the polymorphism in the encoded protein, and (ii) Variant Effect Predictor (VEP) to predict the effect of amino-acid changes through the Sorting Tolerant From Intolerant (SIFT) tool [41]. To highlight variants of the candidate genes with a potential functional impact on the protein function, we selected variants that were classified in terms of their functional consequences as HIGH and MODERATE by the two different software programs. Due to the large number of variants classified as MODERATE, within the moderate missense variants, we selected those predicted to be deleterious through the VEP program [41]. SIFT is an algorithm that predicts whether an amino acid substitution will have a deleterious effect on the protein function [42]. For these potential functionally relevant variants, we later applied a site frequency analysis to identify nucleotides with divergent allele frequency between the two Spanish dairy breeds included in the WGR datasets analyzed here, Spanish Churra (CHU) and Spanish Assaf (ASF). The interpretation of this site frequency analysis also considered the site frequency in the wild sheep ancestor (*O. orientalis*) and Australian Merino samples (MRA) as an example of a non-dairy sheep breed with a sufficient number of samples.

## 3. Results

### 3.1. Genetic Variability and Phylogenetic Relationship between Samples

We analyzed WGR datasets of 43 phenotypically diverse domestic sheep breeds for comparison with 19 Asiatic Mouflon samples (*O. orientalis*) representing their wild ancestor and five wild sheep samples including three *Ovis canadiensis* and two *Ovis dalli* to assess the genetic diversity of the sheep genome regions harboring the list of selected candidate genes under study. The WGR datasets analyzed showed read lengths ranging between 30 and 151 base pairs (bp) and an average number of raw reads per sample of 175,176,421.77 paired reads. After the quality control of raw reads, an average of 16.06% of the reads was eliminated. The number of reads aligned to the reference genome varied between 55,823,646 and 492,895,080, with a mean of accurately mapped reads per sample of 290,256,652.18 and an average of assembly coverage of 94.79%. The median number of duplicated reads per sample was 19,206,080.17, with a range of 1,765,412 to 46,089,858 duplicate reads.

The variant calling analyses performed across the whole genome revealed, after quality filters, a total of 81,053,638 variants across the entire genome, 59.98% of which had been previously described in the database provided by Ensembl (dbSNP release 91). After filtering the regions corresponding to the 24 functional candidate genes considered here, the total number of variants to be further assessed was 20,100, most of which were SNPs (17,037 and 16,960, for SnpEff and VEP, respectively), with lower numbers of insertions (1489 and 1245, for SnpEff and VEP, respectively) and deletions (1574 and 1396, SnpEff and VEP, respectively) (see details in Table 1). These counting differences between these two software packages are due to the different way they consider multiallelic polymorphisms as the SnpEff program considers each polymorphism as a variant and the VEP software contemplates each position contained in a polymorphism as a variant [26,27]. 

Within the candidate gene regions, the rate of variants per base pair (bp) ranged from 26.44; for the *PAEP* gene to 73.25; for the *SCD* gene, with an average ratio of one variant per 54 bp, which was lower than the average variability of one variant per 32 bp estimated across the genome. The number of total and novel variants identified per 500 bp within the considered candidate gene regions is represented in Figure 2. As can be observed, the total number of variants per gene was proportional to the length of the candidate genes, with the *ACACA* gene (with a length of 228,430 bp) involving the higher number of variants. The genes *ACACA*, *XDH*, and *SLC27A6* were those including the highest number of novel variants identified per 500 bp (29, 26, 22, 22 novels per 500 bp variants, respectively). Several insertions and deletions identified in some of the candidate genes (specifically in the *ACACA*, *DGAT1*, *FASN*, and *PAEP* genes) were observed to be coincident among them, having the same reference allele but different length (bp) for the alternate allele. As this observation could indicate that the sequences of the harboring genes are not well-characterized and annotated in the reference genome Oar_3.1, only SNP variants were considered in the further analysis including phylogenetic analysis and the identification of functionally relevant polymorphisms in the candidate genes. The novel variants identified across the studied candidate genes, a total of 4925 from 16,960 SNPs are listed in Appendix A and will be publicly available through the European Variation Archive (EVA) (https://www.ebi.ac.uk/eva/).

A representative sample of 1,331,436 SNPs variants identified across the genome and the 16,960 SNPs variants located within the candidate genes were selected to study the distribution of genetic variability among the samples included in this study. The results of this analysis are graphically presented in a dendrogram plot (Figure 3), preserving the breed code described in Appendix A. The dendrogram shows roughly the genetic differences that arose during the evolution of modern sheep breeds based on the across genome variability (left) and based on the variability identified within the coding regions of the studied candidate genes (right). When focusing on the dendrogram that includes the variability of the entire genome (left), most of the samples are distributed along the phylogenetic tree accordingly to their breed group and their geographical origin region (Figure 1). This agreed with the results reported by Kijas et al. [2] when the Sheep HapMap worldwide sample of sheep breeds was analyzed with a medium-density SNP-chip. Aside from that, the dendrogram based on the variants located within the candidate genes studied here (right) did not allow us to classify the samples by breed, geographical location, or production specialization along the phylogenetic tree, although a clear distinction between wild sheep and domestic sheep could be drawn. The bootstrap values obtained with the dataset including the 1,331,436 SNP variants selected across the genome (represented in Appendix A) showed very high confidence values for the relationship between individuals of the same breed and also for the grouping among breeds of the same geographic regions, which supports the reliability of the phylogenetic tree. 

Likewise, the pairwise distance index between all the analyzed samples was computed using the maximum composite likelihood method based on the variant diversity of the representative sample selected from the variants identified across the genome (Appendix A). The pairwise distance index matrix had an average pairwise distance index of 0.22, ranging from 0.03 to 0.45. The mean distance pairwise index was slightly higher between domestic and wild breeds (0.29) compared to the mean distance pairwise index obtained by comparing distances between different domestic breeds (0.20). The two samples with the highest pairwise distance index with each other were one Australian Merino sample and one *Ovis canadiensis* sample (0.45), and the lowest index was observed between two individuals of the *Ovis dalli* species (0.03). Aside from that, the two groups with the highest pairwise distance index were *Ovis dalli* and *Santa Inês* (0.44), and the lower pairwise distance between groups was found between *Ovis dalli* and *Ovis canadiensis* (0.05). Additionally, regarding the estimation of the nucleotide diversity (π), this parameter was higher in wild sheep samples than in domestic sheep samples when estimated across the whole genome (π = 0.47% and 0.38%, for wild and domestic, respectively) and when estimated for the candidate regions (π = 0.27% and 0.22%, respectively).

Furthermore, the pairwise distance matrix computed based on the 16,960 SNP variants located in the candidate genes showed an average distance of 0.16 (0.02–0.43) [Appendix A]. In the same way, the average divergence index among the wild breeds (0.22) was much higher than that obtained among the domestic breeds (0.13) in the candidate regions. Bootstrap values for the phylogenetic tree based on the candidate gene variants were also high in the roots that classify separately wild and domestic sheep and within breed groups. However, the bootstrap values were markedly reduced in the roots that segregate the domestic breeds (Appendix A). Therefore, this analysis could not correctly classify the breed groups when only variants of candidate genes were used.

### 3.2. Identification of Potential Functionally Relevant Variants in Candidate Genes

The functional annotation carried out with both SnpEff and VEP for the 16,960 SNPs identified in the considered candidate genes through the variant calling analysis applied to the 175 WGR datasets identified five SNP variants determining a HIGH functional impact consequence on the protein. In addition, 203 SNP variants were predicted to cause a MODERATE functional impact consequence, 42 of which were missense variants predicted to be deleterious by the SIFT algorithm implemented in the VEP analysis [42] (Table 1). Considering this classification, a total of 47 functional consequences including those of HIGH impact (5) and the missense deleterious variants (42) were considered as relevant functional variants or variants that are likely to have a direct effect on milk production and composition traits. Twenty of these relevant functional variants are novel as they do not have an rs number in the dbSNP database (https://www.ncbi.nlm.nih.gov/snp/). One of the variants was classified as a stop gained in the *ACACA* gene as it was predicted to cause a codon stop in the coding region that interrupts the elongation of the corresponding protein. 

The 47 mentioned potential functionally relevant variants, which are characterized in Appendix A, were selected to be further explored regarding allelic frequency differences. Hence, for all functional variants, Appendix A provides the estimation of the reference allele frequency for all breeds and also an estimate of the average frequency for domestic and wild sheep. From these estimations, we can observe that a total of the 11 potential functionally relevant variants were only identified in the domestic breeds. Comparing the frequencies between domestic and *O. orientalis*, there was only one polymorphism showing a difference in the allele frequency higher than 0.3 (*ACACA:13029588A>T;* rs589600115), classified as stop gained (Appendix A).

When comparing the allele frequencies of the relevant functional variants between the two dairy breeds considered, Assaf and Churra, we identified two missense deleterious variants with allele frequency differences higher than 0.3 located in the *XDH* and *LALBA* genes (variants highlighted in Appendix A). A complete characterization of these SNPs is presented in Table 2 as potential genetic markers related to the phenotypic differences in milk composition traits between the two considered breeds. On one hand, the alternate allele of the missense deleterious mutation *LALBA:137390760T>C* was close to fixation (0.92), whereas in the Assaf breed, this allele showed a low–moderate allele frequency (0.26). On the other hand, Assaf showed a remarkable higher frequency (0.42) for the alternate allele of the missense deleterious *XDH:92215727C>T* mutation than Churra sheep (0.03), where the reference allele was very close to fixation.

## 4. Discussion

This study exploited the large amount of information provided by WGR datasets from a worldwide sample of sheep breeds to present a deep evaluation on the genetic variability of a list of genes that, due to their known biological function, are considered candidates to explain phenotypic differences for milk composition traits in dairy sheep. One important issue regarding the variant calling analyses reported here is that all the WGR datasets, those obtained from the public SRA repository and those generated by our research group, were processed following the same bioinformatic analysis workflow and were considered jointly for the variant calling analysis. This procedure has been used in previous studies analyzing WGR datasets [35,43,44] and has been proven to prevent biased results regarding the variant calling detection process.

The variants detected by our analysis across the whole sheep genome were used to build a phylogenetic tree that, according to Kijas et al. [2], was able to classify the different breeds and wild species, showing a close relationship between the patterns of ovine genetic variation and geography (Figure 3). Thereby, the highest pairwise distance was observed between one sample of Australian Merino breed and one sample of *Ovis canadiensis*, which is a wild counterpart of the sheep, but not the direct ancestor (*Ovis orientalis*) of the domestic breeds [35]. The individuals that showed the closest relationship in the phylogenetic classification belonged to the same species (*Ovis dalli*). The higher nucleotide diversity observed here in wild vs. domestic sheep has been previously described and can be explained as the consequence of bottlenecks experienced during the domestication of sheep [35,44]. The wild sheep were classified independently from the domestic sheep, considering both the variants identified across the whole genome and the variants included within the studied candidate genes, which is in agreement with the diversity study on the casein genes presented by Luigi-Sierra et al. [44]. However, when the phylogenetic tree was built only considering the variants found in the milk composition candidate genes analyzed here, the domestic sheep samples could not be classified within breeds. According to Luigi-Sierra et al. [44], this may suggest that the set of identified variants identified in the considered candidate genes appear not to have been a target of selection during the breed formation process, as an extensive amount of polymorphisms is shared among the compared domestic breed. Despite that, Naval-Sanchez et al. [35] described a selective sweep located in the coding region of the *DGAT1* gene that could play a role in adaptive changes during domestication or selection. On the other hand, considering that the domestic and the wild sheep samples analyzed here were clearly classified in different nodes based on the genetic variability detected in the considered candidate genes for milk composition, it could be suggested that variants in these genes could have played a role in adaptive changes during the domestication process, as previously suggested by Luigi-Sierra et al. [44] in relation to the casein coding genes. Furthermore, the differences of nucleotide diversity between wild and domestic sheep detailed in this paper agreed with the higher nucleotide diversity previously reported for wild sheep compared with that of domestic sheep by other studies [35,44].

The variant calling bioinformatic pipeline implemented here for 175 WGR datasets from 43 sheep breeds and three wild sheep has allowed for the identification of a total of 47 potential functionally relevant variants located in the candidate genes selected concerning milk composition traits in sheep. Among these variants, 20 were novel. This highlights the value of the analysis presented here for a large number of WGR datasets from a diverse worldwide representation of sheep breeds. In addition to exploiting publicly available information, the results of this analysis build on the sequencing analysis generated for 71 sheep genomes by our research group for four Spanish sheep breeds, with a substantial representation of two dairy sheep breeds, Spanish Assaf (*n* = 19) and Spanish Churra (*n* = 46). This fact is especially relevant if the aim is to analyze genes with a potential influence on dairy traits since public databases have a limited number of these individuals. Nowadays, when genomic selection is possible in some livestock populations, the value of studies identifying genetic variants in candidate genes relays the potential to identify mutations that could be potential causal mutations and also assess the frequency of these mutations in different breeds. This is an important and essential step when designing new SNP arrays that could be used efficiently to improve commercial populations through genomic selection. Different studies have proven that the inclusion of causal mutations in SNP panels used for genomic selection substantially increases the efficiency of this global strategy (reviewed by Xu et al., 2019; Oget et al., 2019).

### 4.1. Variants in Genes Related to Milk Protein Content

Within the coding region of the six candidate genes coding for milk proteins (*CSN1S1*, *CSN2*, *CSN1S2*, *CSN3*, *PAEP*, and *LALBA*), a total of 13 relevant functional SNP variants have been identified. These SNP variants and the estimated frequency for the different breeds analyzed in this study are summarized in Appendix A. Within the functional variants located in the genes related to milk protein content, one was inferred to have a HIGH impact and 12 amino acid substitutions were classified as deleterious. The HIGH impact variant was found on the *CSN1S2* gene, this polymorphism was classified as a splice donor variant and was inferred to cause a disruptive impact on the protein that could affect the gene expression of this gene or could cause an intron retention event and consequently define a novel isoform. One of the missense deleterious variants mentioned was also identified in exon 9 of the *CSN1S2* gene (*CSN1S:85189841G>T*; rs430397133). Of the rest of the missense deleterious variants detected in the genes coding for the casein proteins (Appendix A), two were located on the *CSN1S1* gene, one was on *CSN2*, and three were found in the *CSN3* gene. Among the variants located in the whey protein genes, a total of three deleterious variants were identified in the encoded region of the *LALBA* gene. In contrast, a multi-allelic SNP variant (rs600923112) with three alleles is responsible for the two missense deleterious consequences indicated in Appendix A for the *PAEP* gene. Interestingly, whereas the first of these consequences, related to the substitution *PAEP:3570969T>A* is present uniformly across the different domestic breeds analyzed, the alternative polymorphism *PAEP:3570969T>C* was only found, among the domestic breeds, at very low frequency in Churra and Assaf (0.03 and 0.09, respectively), being also present in the *O. orientalis* samples. A possible association of this functionally relevant variant with the dairy specialization of these two breeds should be studied. 

Polymorphisms in these six candidate genes have classically been considered as potential tools for the selection of dairy ruminants. On one hand, caseins (Cn) constitute 76–83% of the total protein in sheep milk [9,45]. Previous studies had associated the variants in the encoded chain of *CSN1S1* and *CSN1S2* genes with milk yield, protein yield, fat yield, and milk casein content in sheep [9,11]. They were also related to different curd firming times and efficient renneting properties in Sarda goat [46]. Particularly in cow, *CSN2* and *CSN3* genes were significantly associated with the cheese-making traits [1,47]. In sheep, *CSN2* is the gene showing the highest expression level during lactation based on the transcriptomic analysis of somatic milk cells reported by Suárez-Vega et al. [48]. Variants in the *CSN3* gene, which is responsible for stabilizing milk casein micelles, have been associated with protein content and renneting parameters in East Friesian Dairy sheep and Sarda goat, respectively [11,46]. Most of the functional variants found in the coding region of the *CSN3* gene were just present in wild sheep, which is considered monomorphic in domestic sheep [3,20]. Still, our analysis of WGR datasets has identified a missense variant (rs406485755) in one of the Australian Merino samples analyzed. This polymorphism was predicted to cause the p.Arg118His substitution, classified as tolerated (0.32) by our functional analyses. 

On the other hand, the two genes that encode for major whey proteins, α-Lactalbumin (encoded by *LALBA*) and β-Lactoglobulin (*PAEP*, also known as progestagen-associated endometrial protein), both are located in ovine chromosome 3 (OAR3) [49]. García-Gamez et al. [16] identified a quantitative trait nucleotide (QTN) in the *LALBA* gene influencing milk protein and fat percentages in Spanish Churra sheep. Notably, this missense variant (*LALBA:137390760C>T*; p.Val27Ala; rs403176291) was identified here as a potential functional relevant variant, which supports the value and efficiency of the functional filtering process applied in the present study to identify genetic variants with potential biological influence on complex traits. This filtering can help to better understand the genetic architecture of the studied traits by helping to simplify and interpret the huge amount of information generated through standard variant calling analyses of WGR datasets. The *LALBA:137390760C>T* was found in several samples of domestic sheep (Appendix A). The study of García-Gámez et al. [16] shows the direct association of the *LALBA:137390760C* allele with increased milk protein and fat percentage contents. Its high frequency in the Churra breed, compared to the Assaf, agrees with the higher protein and fat contents of the milk from the Churra sheep [6,24]. This may also be related to the better aptitude for mature cheeses of the Churra sheep when compared with the Assaf breed [25,26]. The presence of this polymorphism in the Assaf breed, although at low frequency (0.08), could be considered as a direct genetic marker or through a genomic selection scheme to improve milk composition and cheese-making traits in this breed. Assessment of the potential use of this marker in other dairy breeds should also be considered. The β-Lactoglobulin protein is the main whey protein of ruminant milk and is polymorphic in many sheep breeds [50]. Previous studies have described significant associations among variants found in the *PAEP* gene and protein percentage, fat percentage, clotting time, and curd firming time [17,50,51]. The identification of the *PAEP:3570969C* allele only in the Asian mouflon, Churra, and Assaf breeds, but also in the mouflon, suggest that this mutation has an origin before domestication that has only been observed in breeds with dairy specialization studied in this work. A study involving a larger number of dairy breeds would be necessary to check whether this maintenance is due to reasons of adaptation to milk production or to pure chance.

From the 13 relevant functional variants identified in the genes codifying for milk proteins, four were not previously described (Appendix A). The potential influence of these variants on sheep milk composition, cheese yield, and organoleptic characteristics should be analyzed by future studies. The frequency of these variants in each breed (Appendix A) could be useful to select functional variants to include in the design of SNP-chips aiming for the increase in genomic selection efficiency when applied to dairy breeding programs (e.g., *PAEP:c.500T>C*, only found in Assaf and Churra domestic breeds). One of these novel variants (*CSN3:c.410A>G*) was only present in wild sheep, as previously detailed by Luigi-Sierra et al. [44]. Additionally, a total of 26 QTLs related to milk compositional/functional traits (casein, fat, lactose, and protein percentage) and coagulation properties (curd firmness, curd firming time, and rennet coagulation time) have been identified in a 250 Kb region of these candidate genes in both Churra and Sarda sheep [8,16] (Appendix A).

### 4.2. Variants in Genes Involved in Fatty Acid Metabolism

Within the 18 candidate genes selected in our study on fat metabolism in the sheep mammary gland, a total of 34 relevant functional variants were found, 16 of which were novel variants not included in the dbSNP database. The polymorphisms within these genes, classified as HIGH impact (4) and deleterious missense substitutions (30), and their frequency for the reference allele for each of the studied breeds analyzed, are summarized in Appendix A.

Among the genes related to the fatty acid synthesis and desaturation processes (*ACACA*, *FASN*, and *SCD*), the *ACACA* gene encoding region included a total of four missense deleterious variants and one HIGH impact variant (causing the onset of a stop codon). Aside from that, four deleterious variants were found in the *FASN* gene. In relation to genes related to lipid droplet formation, 12 deleterious variants were found in the *XDH* gene and one in the *BTN1A1* gene. It should also be noted that no relevant functional variants were identified in the genes related to fatty acid activation and intracellular transport.

Regarding the genes associated with the acetate triacylglycerol synthesis (*DGAT1, GPAM*, and *LPIN1*), one missense SNP variant classified as deleterious was identified in the *DGTA1* gene. Furthermore, one SNP variant was classified as HIGH impact in the *GPAM* gene region, which was in the second base region at the 3′ end of the intron, causing a disruptive effect. Finally, a total of three deleterious and two HIGH impact variants were found on the *LPIN1* gene. These HIGH impact variants were located in the 2 bp region at the 3’ end of an intron, which could affect the expression of this gene, causing a disturbing impact on the protein [41], as we have emphasized above. In addition, the SNP located in the *LPIN1* gene (*LPIN1:20554676G>A*) was only identified in the Churra breed genomes. The breed specificity of this variant and the high milk fat content [6,24] and good aptitude of Churra milk for the production of mature cheeses [25,26] suggests future research should focus on the assessment of the potential influence of the *LPIN1:20554676G>A* variant on milk composition traits.

Mammary gland lipid metabolism involves a large number of genes [18]. In this study, we identified a total of 16 novel functional variants in the coding region of five of the 18 candidate genes selected about the milk fatty acid metabolism. The novel polymorphisms were found in the *FASN*, *XDH*, *LPIN1*, *VLDLR*, and *PLIN2* genes (Appendix A). Within the genes related to fatty acid synthesis and desaturation (*ACACA*, *FASN* and *SCD*), the *ACACA* and *FASN* genes are responsible for the fatty acid chain elongation and are both related to the fatty acid synthesis in the mammary gland [17,18]. The high expression of the *FASN* gene and the moderate expression of the *ACACA* gene described in the mammary gland during lactation [21] suggests an essential role of these genes in relation to the synthesis of milk fatty acids. Nine potential functional relevant mutations were identified in the *ACACA* and *FASN* genes. Two of these variants were novel missense mutations in the *FASN* gene, whereas a HIGH impact mutation identified in the *ACACA* gene represented the only *stop_gained* mutation included in our list of potential functional relevant mutations. These results are in contrast with previous studies that had reported a high diversity level of the *ACACA* gene in sheep, but had not identified any non-synonymous mutations [52,53]. This finding illustrates the value of the information generated in this work through the analysis of a large set of WGR samples from a wide range of worldwide sheep breeds. 

Among the genes related to fatty acid metabolism analyzed by Suárez-Vega et al. [21], the *BTN1A1* and *XDH* genes, which encode for butyrophilin and xanthine dehydrogenase, respectively, showed the highest expression levels during lactation. This highlights the importance of lipid droplet formation in the overall process of milk lipid metabolism in sheep. The high expression of the *BTN1A1* gene during lactation has also been described in dairy cows [18], which is in agreement with the crucial role in milk fat secretion suggested previously for this gene by Robenek et al. [54]. Thereby, the 13 relevant functional variants found in the genes *XDH* and *BTN1A1* might affect the function of both proteins and, as a consequence, the lipid droplet formation process [55]. One of the *XDH* missense deleterious variants identified here as a potential functional relevant variant (*XDH:92215727C>T*; rs429850918) showed a certain level of divergence for the frequency of the reference allele between the Assaf and Churra dairy breeds. Whereas the *XDH:92215727C* is segregating in the Assaf breed, this allele is almost fixed in the Churra breed (Table 2). Whether this mutation may explain the higher milk fat contents of Churra sheep compared with Assaf sheep should be further investigated. 

In the three of the candidate genes related to acetate triacylglycerol synthesis [18] coding for diacylglycerol transferase (encoded by *DGAT1* gene), glycerol-3-phosphate acyltransferase mitochondrial (*GPAM* gene), and Lipin 1 (*LPIN1* gene), seven relevant functional variants were identified (Appendix A). One of the novel variants identified in the *LPIN1* gene (*LPIN1_20554676G>A*) was only present in the Assaf breed, which is a highly specialized sheep breed for milk production. This gene has been reported to have a role in the transcriptional regulation of other genes involved in milk lipid synthesis [18]. On the other hand, the *GPAM* gene was the most highly expressed of this group in the sheep mammary gland during lactation [21] whereas the ovine *DGAT1* gene has been associated with a selective sweep caused by the domestication and selection process [35]. Fatty acid synthesis has an important influence on dairy production because it affects the fatty acid composition of milk sheep [52]. Thereby, the relevant functional variants, identified in these genes by the present study, are of interest because they could influence milk composition and cheese-making. From our point of view, the *XDH:92215727C>T* mutation is one of the most promising potential functionally relevant variants highlighted by our study, due to the possible link between the divergence observed for this variant allele frequency between the Assaf and Churra breeds and the known differences in the milk fat contents of these two breeds [21].

Regarding the genes related to the fatty acid import into cells process (*CEL*, *LPL*, and *VLDLR*), one missense deleterious was found in each of the *LDL* and *VLDLR* genes, and two SNPs classified as missense deleterious variants were found in the *PLIN2* gene. In addition, one variant classified as deleterious was identified in the *SLC27A6* gene region. The Lipoprotein lipase (encoded by *LPL* gene) and Very-Low Density Lipoprotein Receptor (*VLDLR* gene) have been previously associated with the function of fatty acid import into cells as the VLDLR is an essential component of LPL activity [18,56]. The deleterious variants described in these genes, which produce missense amino acid substitutions could influence the functionality of the encoded proteins by the *LDL* and *VLDLR* genes. Regarding the other genes related to lipid metabolism, Perilipin-2 (*PLIN2*) is a protein that has been related to the packaging of triglycerides for the secretion of milk lipids in the mammary gland [57], and the *SLC27A6* gene has been found to be upregulated during lactation in dairy cows [18], but was found to have low expression in the sheep mammary gland during lactation [21]. Additionally, when looking for correspondence of previously reported QTL in sheep for milk production traits with the milk fat candidate genes considered here (considering a 250 Kb interval centered on the corresponding gene coding region), we found a total of seven QTLs previously reported within the genomic regions of the *ACACA* and *DGTA1* genes in Altamurana, Gentile di Puglia, and Sarda sheep breeds [53,58] (Appendix A). 

Due to the importance of the candidate genes selected for milk production and composition, the variants described above could explain a proportion of the differences in milk composition and quality between the sheep breeds included in the study. The candidate gene variants detailed might affect the expression and the functionality of the encoded proteins and the respective pathways that are involved. Further research should be conducted to elucidate the genuine impact of the polymorphisms highlighted by the variant filtering workflow implemented here and quantitative and qualitative properties of sheep milk, especially for specific use for high-quality cheese manufacturing. The assessment of allele frequency presented here for the list of potential functionally relevant variants for two dairy breeds might be useful to guide the design of custom SNP chips to be used in genomic selection strategies addressing the genetic improvement of dairy sheep commercial populations.

## 5. Conclusions

The phylogenetic study presented here describes the genetic similarity among domestic and wild sheep. As far as we know, it is the first phylogenetic analysis reported in sheep based on WGR datasets. The pairwise distances provided here might be useful in the design of future genetic studies based on the proximity among the domestic breeds and/or wild sheep. Further research is required to elucidate the full set of genomic regions modified during the process of sheep domestication and genomic selection programs focused on specialized breeds. Furthermore, the functionally relevant variants located in the candidate genes described in this study could be considered as potential markers to increase the efficiency of genetic selection strategies on sheep milk composition traits. Future research is required to confirm the effect of the functionally relevant variants identified in the present study on milk composition and cheese-making traits. 

## Figures and Tables

**Figure 1 animals-10-01542-f001:**
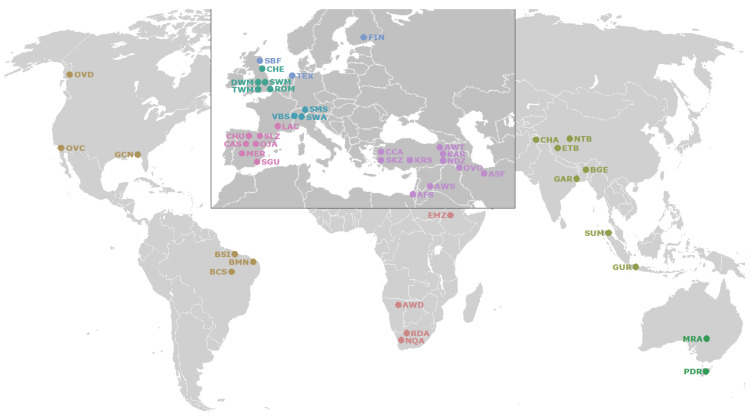
Location breed map. Geographic origin of the samples analyzed by whole genome resequencing (WGR) considered in this study including 151 samples from domestic sheep breeds and 24 samples from wild sheep. Breed and species names and their abbreviations are given in Appendix A. In order to differentiate the geographical origin of the samples analyzed, the geographic area with the highest density of breeds analyzed (Europe and part of Asia) has been zoomed in.

**Figure 2 animals-10-01542-f002:**
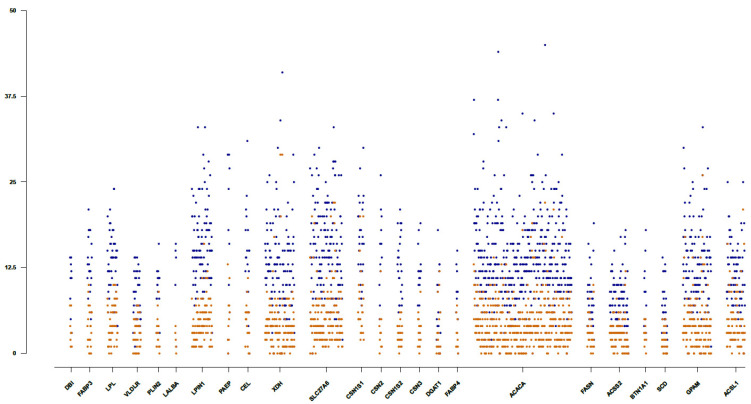
Density of total and novel variants identified in candidate gene regions. Manhattan plot showing the number of variants identified per 500 bp across the selected candidate gene regions. The total and novel variants uncovered in this study are represented in blue and orange colors, respectively.

**Figure 3 animals-10-01542-f003:**
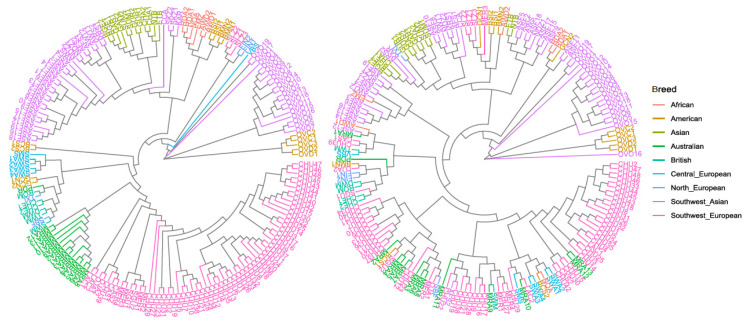
Phylogenetic trees based on the genetic diversity identified across the genome and across and candidate gene regions studied here. Phylogenetic trees were built using the maximum composite likelihood method based on the variant diversity of a representative sample of all the identified SNPs detected across the genome (**left**) and the SNPs detected within the milk candidate genes genomic regions considered in this work (**right**). Phylogenetic trees were drawn using the neighbor-joining method to represent the relationship among all samples analyzed in this study (wild and domestic sheep). The breeds are represented in different colors according to the geographical region to which they belong.

**Table 1 animals-10-01542-t001:** Summary statistics of the annotation analyses performed with the SnpEff and VEP software for the genetic variants identified within the studied candidate gene regions through the analysis of the WGR datasets here analyzed.

Types of Variants According to Different Classification Criteria	Number of Variants within the Candidate Gene Regions	
	Counts SnpEff	Counts VEP
Variants processed	20,100	20,100
SNPs	17,037	16,960
Insertions	1489	1245
Deletions	1574	1396
Indel		422
Substitution		77
Effects by impact (Only for SNPs)		
High	5	5
Moderate	203	203
Low	430	430
Modifier	16,378	16,378
Effects by functional class (Only for SNPs)		
Missense	228	
Nonsense	1	
Silent	424	
SIFT summary (Only for SNPs)		
Deleterious low confidence		11
Tolerated low confidence		19
Deleterious		42
Tolerated		150
Effects by type (Only for SNPs)		
3 prime UTR	113	113
5 prime UTR	59	59
Downstream gene	651	651
Frameshift	0	0
Inframe deletion	0	0
Inframe insertion	0	0
Intergenic region	0	0
Intron	18,579	18,574
Missense	228	228
Non coding transcript exon	12	12
Non coding transcript	137	137
Splice acceptor	4	4
Splice donor	1	1
Splice region	97	96
Start lost	0	0
Stop gained	1	1
Stop lost	0	0
Stop retained	0	0
Synonymous	424	424
Upstream gene	729	729

**Table 2 animals-10-01542-t002:** Characterization of the two potential functionally relevant variants identified in the candidate genes considered in this study showing allele frequency divergence between Churra and Assaf breeds.

Features	*XDH:92215727C>T*	*LALBA:137390760T>C*
Chromosome	3	3
Position (base pairs)	92,215,727	137,390,760
dbSNP ID	rs429850918	rs403176291
Reference Allele	C	T
Alternate Allele	T	C
GeneSymbol	*XDH*	*LALBA*
Variant	missense	missense
BioType	protein coding	protein coding
Functional impact (ensemblVEP_Oarv3.1)	*MODERATE*	*MODERATE*
Functional impact (SIFT_Oarv3.1)	deleterious (0)	deleterious (0.02)
Positions in coding sequence	*ENSOART00000011926.1:c.1840C>T*	*ENSOART00000020933.1:c.80T>C*
Codon change	Cgg/Tgg	gTg/gCg
Amino acid substitution	Arg/Trp	Val/Ala
Assaf Genotypes (Frequency)	C (0.58), T (0.42)	T (0.92), C (0.08)
Churra Genotypes (Frequency)	C (0.97), T (0.03)	T (0.26), C (0.74)

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
