# Peer review of "Analysis of Whole Genome Resequencing Datasets from a Worldwide Sample of Sheep Breeds to Identify Potential Causal Mutations Influencing Milk Composition Traits"

_animals, 2020, doi:10.3390/ani10091542_

Round 1
Reviewer 1 Report
The author analyzed WGR datasets from a worldwide sample of sheep breeds, including domestic sheep breeds and wild sheep, which were already used in previous studies and identified the number of variants detected across the genome and within the milk candidate gene previously proved to be related to milk composition.
However, the paper is basically a description of whole genome resequencing datasets and doesn’t carry any interesting conclusions or novel scientific results. We do not get any new information that can guide genetic selection strategies on sheep milk composition traits, as the author wrote that most of the domestic sheep analysed WGR datasets were publicly available from two different projects at the Sequence Read Archive (SRA) repository, including candidate genes considered. Furthermore, I believe the data are not robust enough to deduce any of the stated conclusions,For example –“Genetic variability and phylogenetic relationship between samples, and the functionally relevant variants located in the candidate genes described in this study”. there is no certain evidence showing the variants are relevant to sheep milk composition traits.
In conclusion, I do not think this manuscript is worth to be published in a scientific journal as an article
Author Response
Reviewer 1
The author analyzed WGR datasets from a worldwide sample of sheep breeds, including domestic sheep breeds and wild sheep, which were already used in previous studies and identified the number of variants detected across the genome and within the milk candidate gene previously proved to be related to milk composition.
However, the paper is basically a description of whole genome resequencing datasets and doesn’t carry any interesting conclusions or novel scientific results. We do not get any new information that can guide genetic selection strategies on sheep milk composition traits, as the author wrote that most of the domestic sheep analyzed WGR datasets were publicly available from two different projects at the Sequence Read Archive (SRA) repository, including candidate genes considered.
Au: We understand that if the Reviewer has this feeling about our work, we should make a significant effort on highlighting the value of the work and results presented here. Hence, the comment of the Reviewer can be of great value to improve our manuscript. Indeed, several (104) of the 175 WGR datasets analyzed in our study are from publically available SRA. Still, we think the Reviewer should take into account the value of the 71 WGR datasets generated by our research group for two dairy sheep breeds, Spanish Assaf and Spanish Churra sheep. Considering the limited representation of dairy breeds in the SRA available datasets (only two Lacaune samples among the European breeds), the datasets generated by our group provide valuable information that can be specifically of interest to the dairy sheep research community. This point of value is highlighted now in lines 442-445 of Discussion.
Also, we have performed a joint analysis of all the WGR datasets, which is a major challenge for the analysis of this kind of datasets. On this regard, the Reviewer should be aware that our study has involved the analysis from scratch for all the considered 175 WGR datasets, not only for those generated by our group. This approach allowed us to increase the number of samples to consider in this study when an initial proportion of them had already been analyzed and has generated files that can be analyzed in the future with additional samples. In addition, this approach increases the quality of the variant quality analysis as several samples were jointly analyzed. Also, the use of version 4.0 of the GATK standard software for variant calling has increased our ability to identify variants with appropriate technical support due to the improvements of this version of the software. The technical value of the bioinformatic approach implemented in our study is highlighted in lines 399-404.
Apart from that, we would like to highlight the interest that the identification of polymorphisms in candidate genes for milk production has had for several years. A list of examples of representative studies in this field is provided below, from the classic candidate gene studies in the 1990s (e.g. Moioli et al., 1998; reviewed by Moioli et al. 2007) to the most recent publications exploiting novel technologies such as RNA-Seq (e.g. Suarez-Vega et al., 2017) or whole genome sequencing (Luigi-Sierra et al., 2020). Such a large list of studies supports the interest of our study and highlights the great opportunity that the analysis of this worldwide set of ovine WGR datasets offers to identify novel mutations in candidate genes. On this regard, our study has identified a total of 32,435,762 novel variants across the genome, from the total 81,053,638 (see line 266-267). From them, 4,925 novel mutations were distributed within the 24 studied candidate genes. The finding of these novel mutations is highlighted now in lines 296-298.
Nowadays, when genomic selection is possible in some livestock populations, the value of these studies identifying genetic variants in candidate genes relays on the potential to identify mutations that could be causal mutations and assess the frequency of these mutations in different breeds. This an essential step before designing new versions of the SNP-arrays that could be used to improve the accuracy of genomic selection. This justifies the interest of this study about guiding genetic selection strategies on sheep milk composition traits. We have added a general comment about this in lines 693-696.
On this regard, we think the information provided in Table S6 about the allele frequency of the potential functional relevant variants for all the 43 breeds analyzed is very interesting, especially for the Churra and Assaf Spanish dairy breeds, for which a representative number of samples have been analyzed and for which the information related to the studied candidate genes for milk composition traits may be of higher interest. This information has a direct practical use, as mutations that are monomorphic in these breeds should be discarded during the selection of SNPs to be included in a custom SNP-chip addressing genetic improvement of commercial populations of these breeds. As said before, we think this justifies that the information generated in this study can be used to guide genetic selection strategies on sheep milk composition traits (highlighted now in lines 445-452;693-696).
We acknowledge that, perhaps, we had not previously appropriately highlighted the value of the information generated for the novel variants detected by exploiting the information of the broad set of WGR samples analyzed here. To solve this limitation, we provide now information about the 4,925 novel variants identified for these genes in a new Table S3, which will be submitted to the EVA database variant repository, as indicated now in lines 296-298. At this moment we do not have an EVA accession number because this repository requires a published publication to link the submitted dataset of novel variants. Because of that, if this paper is accepted, we will link the submitted novel variants to this publication. Note that 20 of these novel variants in the candidate genes are also included in updated Table S6 as they were identified as potential functional relevant variants.
As one example of the value of the novel variants identified in the candidate genes, we can cite, for instance, the missense variants identified in the ACACA gene, where previous studies analyzing several breeds had not identified any missense variant (Garcia-Fernandez et al. 2010; Moioli et al., 2013). We have added a comment about this example in lines 578-584.
Moioli, B.; Pilla, F.; Tripaldi, C. Detection of milk protein genetic polymorphisms in order to improve dairy traits in sheep and goats: a review. Small Rumin. Res. 1998, 27, 185–195.
Moioli, B.; D’Andrea, M.; Pilla, F. Candidate genes affecting sheep and goat milk quality. Small Rumin. Res. 2007, 68, 179–192.
Noce A, Pazzola M, Dettori ML, Amills M, Castelló A, Cecchinato A, et al. Variations at regulatory regions of the milk protein genes are associated with milk traits and coagulation properties in the Sarda sheep. Anim Genet [Internet]. 2016 Dec 1 [cited 2019 Sep 25];47(6):717–26. Available from: http://doi.wiley.com/10.1111/age.12474
García-Fernández, M.; Gutiérrez-Gil, B.; García-Gámez, E.; Sánchez, J.P.; Arranz, J.J. The identification of QTL that affect the fatty acid composition of milk on sheep chromosome 11. Anim. Genet. 2010, 41, 324–328.
Moioli, B.; Scatà, M.C.; De Matteis, G.; Annicchiarico, G.; Catillo, G.; Napolitano, F. The ACACA gene is a potential candidate gene for fat content in sheep milk. Anim. Genet. 2013, 44, 601–603.
Suárez-Vega A, Gutiérrez-Gil B, Klopp C, Tosser-Klopp G, Arranz JJ. Variant discovery in the sheep milk transcriptome using RNA sequencing. BMC Genomics [Internet]. 2017;18(1):170. Available from: http://bmcgenomics.biomedcentral.com/articles/10.1186/s12864-017-3581-1
Luigi-Sierra MG, Mármol-Sánchez E, Amills M. Comparing the diversity of the casein genes in the Asian mouflon and domestic sheep. Anim Genet [Internet]. 2020; Available from: http://doi.wiley.com/10.1111/age.12937
Furthermore, I believe the data are not robust enough to deduce any of the stated conclusions,For example –“Genetic variability and phylogenetic relationship between samples, and the functionally relevant variants located in the candidate genes described in this study”. there is no certain evidence showing the variants are relevant to sheep milk composition traits.
In conclusion, I do not think this manuscript is worth to be published in a scientific journal as an article
Au: Regarding this criticism of the Reviewer, we would like to highlight that nowadays several studies have exploited whole-genome sequencing datasets to identify and study genetic diversity in different livestock and this is a valuable approach to extract global information from animal species genomes searching for selection signatures (See the list copied below). From that list, we would like to highlight the study presented by Naval-Sanchez et al., (2018) which show similar analyses than those presented here concerning genetic variability and phylogenetic relationship but considering all the variability detected across the genome. In our study, we have added new WGR datasets than those previously analyzed by Naval-Sanchez et al. (2018) and presented global results of variability and phylogenetic relationships based on variants across the whole genome and variants located in milk candidate genes. Our findings and conclusions are similar to those presented by Naval-Sanchez et al. (2108), and in addition provide a new different view by focusing on genes of interest for milk production, supported by the analysis of dairy sheep breeds which had not been previously analyzed (the Assaf and Churra WGR datasets). Also, the research approach, results and conclusions presented here are similar to those presented by Luigi-Sierra et al. (2020) which exploited WGR datasets to study the genetic variability of the casein coding genes. Hence, our paper adds valuable information to that publication considering in addition to the casein genes, the genes coding for the whey milk proteins and also a large number of candidate genes related to the milk fat synthesis. Discussion of our results with these two studies is provided in the Discussion section of the manuscript.
Naval-Sanchez M, Nguyen Q, McWilliam S, Porto-Neto LR, Tellam R, Vuocolo T, et al. Sheep genome functional annotation reveals proximal regulatory elements contributed to the evolution of modern breeds. Nat Commun [Internet]. 2018;9(1):1–13. Available from: http://dx.doi.org/10.1038/s41467-017-02809-1
Stothard P, Choi JW, Basu U, Sumner-Thomson JM, Meng Y, Liao X, et al. Whole genome resequencing of black Angus and Holstein cattle for SNP and CNV discovery. BMC Genomics [Internet]. 2011 Nov 15 [cited 2020 Aug 17];12(1):1–14. Available from: https://link.springer.com/articles/10.1186/1471-2164-12-559
Luigi-Sierra MG, Mármol-Sánchez E, Amills M. Comparing the diversity of the casein genes in the Asian mouflon and domestic sheep. Anim Genet [Internet]. 2020; Available from: http://doi.wiley.com/10.1111/age.12937
Mei C, Wang H, Zhu W, Wang H, Cheng G, Qu K, et al. Whole-genome sequencing of the endangered bovine species Gayal (Bos frontalis) provides new insights into its genetic features. Sci Rep [Internet]. 2016 Jan 25 [cited 2020 Aug 17];6(1):1–8. Available from: https://www.nature.com/articles/srep19787.
About the comment “there is no certain evidence showing the variants are relevant to sheep milk composition traits”, we think we have been careful about that across all the manuscript, but according to the Reviewer´s comment we have tried to be even more cautions. See below.
The title was already referring to “potential” causal mutations (line 4). In methods, when commenting about the type of mutations we are going to study in detail, we had already said: “with a potential functional impact on the protein function” (line 239-240). Those variants were previously referred across the manuscript as “functionally relevant variants”. But considering the point highlighted here by the Reviewer, we have added the term “potential” to all the references to “functional relevant variants”. See changes made in lines 245, 385, 399, 403, 425, 469, 533, 609, 612, 615, 626, 651, 679. Hence, we think that now we are very prudent about the fact that the potential effect of the mutations highlighted by our bioinformatic filtering process in relation to milk production/composition traits should be confirmed by further studies, as we had previously commented in lines 690-693 and 701-706. However, we think that the fact that our bioinformatic filtering process has identified the LALBA:137390760T>C mutation as a potential relevant variant and the previous report of this mutation as the causal mutation of a QTL for the milk protein and fat contents (Garcia-Gámez et al. 2012) supports the validity of the approach presented in this work.
García-Gámez, E.; Gutiérrez-Gil, B.; Sahana, G.; Sánchez, J.-P.; Bayón, Y.; Arranz, J.-J.; Jiang, L.; Liu, J.; Sun, D.; Ma, P.; et al. GWA Analysis for Milk Production Traits in Dairy Sheep and Genetic Support for a QTN Influencing Milk Protein Percentage in the LALBA Gene. PLoS One 2012, 7, e47782.
Finally, we would like to thank the Reviewer for providing us with his perspective on the article. However, we would like to indicate our cordial disagreement with the Reviewer opinion. As we tried to argue in our response, we think that our manuscript and those referenced in this response are an important part of the scientific advance in animal genomics. The analysis of the existing variability in the genome of domestic animals from a basic point of view constitutes a fundamental tool of animal science that can provide tools to be applied in efficient genomic selection processes and also in the knowledge of the genetic architecture of complex animal phenotypes. We hope that the changes made to address the Reviewer´s comment have helped to highlight the value of the work presented here and can make our manuscript suitable for publication.
Reviewer 2 Report
The manuscript “Analysis of whole-genome resequencing datasets from a worldwide sample of sheep breeds to identify potential causal mutations influencing milk composition traits” is very well written and interesting. It will have a great impact on the genetics of milk production in sheep.
All methods and results are well presented, and the discussion is deep and fruitful while the introduction is concise and informative.
I have several small comments.
Line 31- 32: might be written as candidate genes for milk compositions
Line 32: you might need to define SNP, even we already know what it is.
Line 32: might remove variants after SNPs because SNPs are variants
Line 37: I would prefer to change XDH_92215727C>T to XDH:92215727C>T or you can write _92215727C>T in XDH gene, or you can use the rs number rs429850918 for it. Same suggestion for other SNP
Line 157: I believe that it will be better if the text in the map can be readable
Figure 3. The authors might use the horizontal alignment so have more space for each figure.
Table 1. Did authors check different subclass in missense variant groups? How are about other groups in the variant classification (https://m.ensembl.org/info/genome/variation/prediction/predicted_data.html)
Do the authors have any information about the percentage of the phenotypic variance for milk composition traits explained variance selected genes?
Is it possible to develop an SNP panel for genetic selections based on the list of the selected candidate genes?
Line 501: base pair can write as bp
Author Response
Reviewer 2
The manuscript “Analysis of whole-genome resequencing datasets from a worldwide sample of sheep breeds to identify potential causal mutations influencing milk composition traits” is very well written and interesting. It will have a great impact on the genetics of milk production in sheep.
All methods and results are well presented, and the discussion is deep and fruitful while the introduction is concise and informative.
Au: We would like to thank the Reviewer about the comments on our manuscript and about the detailed, constructive corrections presented below, which undoubtedly improve our manuscript and the interest shown in our work.
I have several small comments.
Line 31- 32: might be written as candidate genes for milk compositions
Au: Corrected, we have added: “candidate genes for milk composition” (line: 31-32).
Line 32: you might need to define SNP, even we already know what it is.
Au: Following the Reviewer’s suggestion, we have defined this acronym in the abstract and the first time that it is used in the manuscript (line: 33 and 224).
Line 32: might remove variants after SNPs because SNPs are variants
Au: Removed (line: 33-34).
Line 37: I would prefer to change XDH_92215727C>T to XDH:92215727C>T or you can write _92215727C>T in XDH gene, or you can use the rs number rs429850918 for it. Same suggestion for other SNP
Au: Following the Reviewer’s suggestion, we have changed the underscore to a colon in specific this line (line: 38) and for all the variant names mentioned across the manuscript.
Line 157: I believe that it will be better if the text in the map can be readable
Au: Following the Reviewer’s suggestion, we have made a zoom on the geographical regions, including the largest groups of studied sheep breeds and have increased the font size of the breed names. We provide now a high-resolution image of this updated figure and hope the zoom approach helps to make the text of the map more readable now.
Figure 3. The authors might use the horizontal alignment so have more space for each figure.
Au: The horizontal alignment suggested by the Reviewer would take one page for each figure, as in Supp. Figure S1 and S2. Following the Reviewer´s comment, we present now a horizontal legend, so the available space for each figure is increased.
Table 1. Did authors check different subclass in missense variant groups? How are about other groups in the variant classification (https://m.ensembl.org/info/genome/variation/prediction/predicted_data.html)
Au: For simplicity, we had not added information about the classification referred by the Reviewer. But considering that the Reviewer requests this information we have added the classification for the variants included in coding regions into Table 1 (“Effects by type” section). We are not sure about the subclasses mentioned by the Reviewers in the missense variant group, as the classification indicated refers to several classes on the same hierarchical level than missense. In any case, we already had provided the classification of the missense variants according to the SIFT analysis results (Table 1: “SIFT summary” section).
Do the authors have any information about the percentage of the phenotypic variance for milk composition traits explained variance selected genes?
Au: The Reviewer points out an interesting issue. However, because for some of the selected genes studies in different breeds have shown different results, it isn't very easy to estimate the proportion of the variance explained by the selected candidate genes. In addition, most of the associations reported for these genes concerning milk compositions traits have been reported in associations studies performed on a limited number of animals, and the estimated effects may be overestimated. For instance, the GWAS performed in Churra sheep by our group, reported by García-Gámez et al. (2012), estimated for the mutation LALBA:137390760T>C an allele substitution effect of 0.470 (in phenotypic standard deviations of the trait: yield deviation for milk protein percentage). It is evident that this effect is overestimated (The winner's curse effect: doi: 10.1371/journal.pgen.1006916). In our experience, this mutation which has a very low frequency in the Assaf breed does not explain such proportion of the phenotypic variance in this breed. Hence, we think that only by performing large studies with many different breeds for a list of potential causal variants we could estimate the percentage of the phenotypic variance that the selected candidate genes explain for milk composition traits in sheep. Hence, our study would be one step forward towards gaining knowledge on the interesting point the Reviewer has indicated here.
Is it possible to develop an SNP panel for genetic selections based on the list of the selected candidate genes?
Au: Yes, considering that the functionally relevant variants highlighted by the present work are SNPs, these variants could be included in a new or be added in a previously designed custom SNP panel. The inclusion of these variants in an SNP panel and its genotyping in commercial populations with available phenotypic data would help to elucidate their effect on milk production and composition traits and also concerning cheese-making traits. Besides, the SNP panel could be used in genomic selection programmes applied in dairy sheep. Different studies have proved that the inclusion of causal mutations in SNP panels used for genomic selection substantially increases the efficiency of this global strategy (reviewed by Xu et al., 2019 doi: 10.1016/j.xplc.2019.100005; Oget et al., 2019, doi: 10.1186/s12864-019-6068-4). This supports the value of studies like the one here presented. In addition, the information of allele frequency presented here for the functionally relevant mutations for the different breeds, especially for Churra and Assaf, is an important and essential step when designing new SNP arrays that could be used efficiently to improve commercial populations through genomic selection. The value of this study in relation to this point is highlighted now in lines 445-452 and lines 693-696.
Line 501: base pair can write as bp
Au: Corrected (line: 562).
Reviewer 3 Report
Reviewer #: Manuscript ID animals-886865
General Comments:
The manuscript by Marina and collaborators entitled “Analysis of whole genome resequencing datasets from a worldwide sample of sheep breeds to identify potential causal mutations influencing milk composition traits” exploited the information derived from a total of 175 whole-genome resequencing (WGR) datasets from 43 domestic sheep breeds and three wild sheep to evaluate the genetic diversity of 24 milk composition candidate genes.
This interesting study provide a novel vision of how the variants found in certain candidate genes participate on sheep milk composition traits. Some of the variants found in this study in these selected genes could be participating in the known differences in milk fat and proteins contents between the Assaf and Churra Spanish breeds.
The functionally relevant variants identified in the candidate genes studied in this work could be suggested as potential markers to increase the efficiency of genetic selection strategies on sheep milk composition traits. The experimental design used in this study is robust and consistent, the statistical analyses are rigorous and have been performed consistently and the manuscript has been written in correct English without any apparent error. However, some modifications and changes are recommended before it will be published.
Materials and Methods:
L.121-153: In these paragraphs, the number of datasets used is quite confusing, although it is clarified in the supplementary table indicated; it is true that many times when works are consulted the supplementary material is not reviewed. I would advise you to modify these paragraphs and indicate the ten Australian merinos that come from a project other than PRJNA160933, even if it is from the same data repository.
Discussion:
L.393-396: Please indicate a reference that supports this paragraph or rewrite it in another way.
L.410: Please indicate the six candidate genes in parentheses.
L.516-518: Please rewrite this sentence is speculative
Author Response
Reviewer 3
General Comments:
The manuscript by Marina and collaborators entitled “Analysis of whole genome resequencing datasets from a worldwide sample of sheep breeds to identify potential causal mutations influencing milk composition traits” exploited the information derived from a total of 175 whole-genome resequencing (WGR) datasets from 43 domestic sheep breeds and three wild sheep to evaluate the genetic diversity of 24 milk composition candidate genes.
This interesting study provide a novel vision of how the variants found in certain candidate genes participate on sheep milk composition traits. Some of the variants found in this study in these selected genes could be participating in the known differences in milk fat and proteins contents between the Assaf and Churra Spanish breeds.
The functionally relevant variants identified in the candidate genes studied in this work could be suggested as potential markers to increase the efficiency of genetic selection strategies on sheep milk composition traits. The experimental design used in this study is robust and consistent, the statistical analyses are rigorous and have been performed consistently and the manuscript has been written in correct English without any apparent error. However, some modifications and changes are recommended before it will be published.
Au: We would like to thank the Reviewer’s comments, the interest in this study and the suggestions provided, which have helped us to improve our manuscript.
Materials and Methods:
L.121-153: In these paragraphs, the number of datasets used is quite confusing, although it is clarified in the supplementary table indicated; it is true that many times when works are consulted the supplementary material is not reviewed. I would advise you to modify these paragraphs and indicate the ten Australian merinos that come from a project other than PRJNA160933, even if it is from the same data repository.
Au: Following the Reviewer’s suggestion, we have identified some errors in the figures given in the text, and we have corrected them accordingly to Table S1 (lines 127-136). We have also rewritten this part of the document and tried to clarify how many samples were obtained from the two different SRA projects publically available and explained how many of them are within the domestic and wild groups (lines 127-136). In addition, we now indicate the SRA references of the WGR projects, so the reader does not need to go to the supplementary material to understand the Australian Merino samples come from the PRJNA325682 project (line: 134). Also, because we agree with the Reviewer that having all the SRA project, references in the text is a great help for the reader.
Discussion:
L.393-396: Please indicate a reference that supports this paragraph or rewrite it in another way.
Au: Thanks for the comment. We have rewritten the sentence and added an appropriate reference (line: 428-432; Text starting as: “On the other hand, considering that the domestic and the wild…”.
L.410: Please indicate the six candidate genes in parentheses.
Au: Following the Reviewer’s suggestion, the six candidate genes have been added (line: 454-455).
L.516-518: Please rewrite this sentence is speculative
Au: Thanks for the comment. We have removed the speculative sentence which was not informative, and we have added a short discussion about previous articles studying the variability of the ACACA gene, which put into value the results presented here (line: 577-584).
Round 2
Reviewer 1 Report
We knew that 71 WGR datasets generated by the author’s group for two dairy sheep breeds, which is valuable and enriched the genomic data information of sheep. However, there is no information of sheep related traits and genome association analysis. Therefore, based on the results author presented, there is no certain evidence showing the variants are relevant to sheep milk composition traits.
The study author presented is only partly similar with several studies listed in the response letter. For example, in the study about sheep genome functional annotation (Marina Naval-Sanchez et al.,2018), in addition to the analysis mentioned in your study, Marina Naval-Sanchez et al showed substantial amount of work, they also validated sheep genome functional annotation using experimental ChIP-Seq of sheep tissue in their study.
In conclusion, there is still lack of evidence showing “potential” variants. I do not think this manuscript is worth to be published in Animals at this stage.